# Balancing Functionality and Printability: High-Loading Polymer Resins for Direct Ink Writing

**DOI:** 10.3390/polym14214661

**Published:** 2022-11-01

**Authors:** Shelbie A. Legett, Xavier Torres, Andrew M. Schmalzer, Adam Pacheco, John R. Stockdale, Samantha Talley, Tom Robison, Andrea Labouriau

**Affiliations:** 1Los Alamos National Laboratory, Los Alamos, NM 87545, USA; 2Kansas City National Security Campus Managed by Honeywell Federal Manufacturing & Technologies LLC, Kansas City, MO 64147, USA

**Keywords:** direct ink writing, 3D printability, rheology, thermomechanical properties, lattice structure, functional resins, powdered fillers, PDMS

## Abstract

Although direct ink writing (DIW) allows the rapid fabrication of unique 3D printed objects, the resins—or “inks”—available for this technique are in short supply and often offer little functionality, leading to the development of new, custom inks. However, when creating new inks, the ability of the ink to lead to a successful print, or the “printability,” must be considered. Thus, this work examined the effect of filler composition/concentration, printing parameters, and lattice structure on the printability of new polysiloxane inks incorporating high concentrations (50–70 wt%) of metallic and ceramic fillers as well as emulsions. Results suggest that strut diameter and spacing ratio have the most influence on the printability of DIW materials and that the printability of silica- and metal-filled inks is more predictable than ceramic-filled inks. Additionally, higher filler loadings and SC geometries led to stiffer printed parts than lower loadings and FCT geometries, and metal-filled inks were more thermally stable than ceramic-filled inks. The findings in this work provide important insights into the tradeoffs associated with the development of unique and/or multifunctional DIW inks, printability, and the final material’s performance.

## 1. Introduction

The key benefit of additive manufacturing is the ability to quickly produce three-dimensional objects for a wide range of applications: from replacement parts for common household items to works of art and entirely new devices designed to solve unique problems. A part can go from an idea to a physical object in a matter of days or even hours rather than months or years. Along with the ability to create tailor-made items for specific applications, 3D printing also allows one to choose the materials, or feedstock, used to manufacture them. Feedstock for 3D printing comes in many forms depending on the type of 3D printing technology. For instance, fused filament fabrication (FFF), uses filaments made of thermoplastic materials that may incorporate fillers to give various desirable characteristics (i.e., mechanical strength, thermal conductivity, or color-changing) to 3D printed parts. In the case of FFF, many of these functional filaments are commercially available, while others are made in-house to suite specific needs. On the other hand, feedstocks for direct ink write (DIW) 3D printing are made up of shear-thinning fluids, or “inks”, and, while many labs make their own custom inks to fulfill specific needs, there are many fewer commercially available options compared to FFF filaments [1,2,3,4,5,6,7,8,9,10,11].

The ability to use polymer resins as inks for DIW depends greatly on their rheological properties such as equilibrium storage modulus (the stress at which a material behaves like a viscoelastic solid) and yield stress (the stress at which a material flows). For DIW printing, the resin must be a shear-thinning yield stress fluid, where stress exceeding the yield stress is reached within the printing nozzle and then relieved after ink deposition. By applying greater stress during extrusion and removing the stress following deposition, a shear-thinning ink can flow from the nozzle and then regain the solidity needed to support itself once deposited on the build plate [12,13,14,15,16]. Achieving these rheological properties when developing a new resin is difficult enough, but including functional fillers in a DIW ink adds further difficulty. When creating custom feedstock to imbue 3D printed parts with specific functionalities, the goal is to add as much of the substances granting functionality to the material as possible while still being able to print the desired objects accurately and precisely. This can be a tricky endeavor since the foundations of most 3D printing feedstocks are polymers, while the added functional materials are usually metals, ceramics, or other powders that may detract from optimal mechanical properties like flexibility, flow, and tensile strength that make the polymers ideal materials for 3D printing feedstock [17,18]. For DIW printing in particular, adding functional fillers to polydimethylsiloxane (PDMS)-based inks can result in nozzle clogs, discontinuous flow onto the print bed, and increased sagging in areas of a print where bridging occurs leading to deviations in desired print dimensions [18,19]. The ability of an ink to create an object accurately and precisely as determined by the print code will be referred to as “printability”.

Previous work by this group has examined the relationship between single metallic and ceramic filler concentrations up to 50 wt%, or multi-filler concentrations up to 70 wt%, and the resulting printability and mechanical properties of face-centered tetragonal (FCT) or square-centered (SC) lattice structures [18,19]. This study builds upon previous work and analyzes the printability limits and thermomechanical characteristics of DIW inks containing 50–70 wt% metallic and ceramic fillers printed in FCT and SC lattice geometries.

## 2. Materials and Methods

### 2.1. Materials

The inks created in this study consist of a siloxane and fumed silica base similar to that of Brounstein et al. [19]. Siloxanes used in this study include vinyl-terminated (4–6% diphenylsiloxane)-dimethylsiloxane copolymer (Gelest PDV-541) and trimethylsiloxy-terminated methylhydrosiloxane-dimethylsiloxane copolymer (Gelest HMS-301) (Gelest, Inc., Morrisville, PA, USA). To prevent premature curing of the inks, we used 1-ethynyl-1-cyclohexanol (ETCH; 99%, Sigma Aldrich) (Millipore Sigma, St. Louis, MO, USA). A high-temperature platinum catalyst (Gelest SIP 6829.2; platinum carbonyl cyclovinylmethylsiloxane complex; 1.85–2.1% Pt in cyclomethyl vinyl siloxanes) and a mid-temperature platinum catalyst (Gelest SIP 6832.2; platinum-cyclovinylmethyl-siloxane complex; 2% Pt in cyclomethylvinylsiloxanes) were used to induce crosslinking. Fillers included an OH-functionalized fumed silica (Evonik Aerosil 300, Evonik Industries AG, Essen, Germany), a PDMS-functionalized silica (CAB-O-SIL TS-720; Cabot Corporation, Boston, MA, USA), powdered tungsten metal (99.9%, American Elements and 99.9%, Sigma Aldrich) (American Elements, Los Angeles, CA, USA), powdered tungsten oxide (99.9%, American Elements and ≥99%, Sigma Aldrich), powdered gadolinium oxide (99.9%; American Elements), and a surfactant (TWEEN 60, Sigma Aldrich).

### 2.2. Resin Formulation and Printing Parameters

The base of all resins created for this study consisted of a 9:1 ratio of PDV-541 to HMS-301 and <1 wt% ETCH. Aerosil 300 (A300) or CAB-O-SIL TS-720 (TS-720) was added to all inks to enhance shear-thinning characteristics, with A300/A300 Emulsion and TS-720 inks containing 10 wt% A300 and 15 wt% TS-720, respectively, and all other inks, with the exception of the Emulsion ink, containing 4.5 wt% A300 (Table 1). All metal and ceramic fillers (W, WO_3_, and Gd_2_O_3_) were sieved (Gilson Company, Inc., Lewis Center, OH, USA) to a maximum particle size of 53 µm before adding them to base resins to create 50, 60, and 70 wt% metal/ceramic-filled inks (Table 1). Emulsion inks (Emulsion and A300 Emulsion) were created in an attempt to promote the formation of pores within the ink; for these inks, 15 wt% 18.2 MΩ water and <1 wt% TWEEN 60 were added to the base resin formulations. All components were combined in disposable plastic cups and mixed with a THINKY planetary vacuum mixer (ARV-310, THINKY USA INC, Verdugo, CA, USA) under ambient pressure. Finally, 0.1 wt% platinum catalyst was mixed into each ink before sealing the inks in plastic bags and storing them at 4 °C until ready for printing. The compositions and densities for all inks created for this study can be seen in Table 1. Early attempts to print with the inks shortly after preparation failed because their viscosities were too low. To prevent this, inks were stored in a refrigerator at least overnight before printing, which promoted polymer/silica interactions and network formation leading to more viscous resins.

Resins were loaded into 25 mL stainless steel syringes and centrifuged at 2000 rpm for 1–2 min to concentrate the resin at the bottom of the syringe and remove any air bubbles. The filled syringes were attached to a Hyrel EMO-XT print head (Hyrel 3D, Atlanta, GA, USA) and then connected to a Hyrel System 30 3D printer. Inks were extruded onto the build plate at room temperature using 250 µm or 410 µm diameter plastic luer-lock nozzles (Nordson EFD Precision Tips, Nordson Corporation, Westlake, OH, USA). The printer was controlled using Repetrel software (Hyrel 3D) and custom G-code which specified the lattice geometry (FCT or SC) and center-to-center spacing between struts (500 µm, 750 µm, 1000 µm, or 1500 µm). Print settings (shown in Table 2) were determined empirically through observations made while printing inks with various rheologies, filler concentrations, and particle shapes/sizes. For example, nearly all inks with filler concentrations > 50 wt% quickly and consistently clogged 250 µm nozzles, leading to the use of 410 µm nozzles for these inks. Layer heights (225 µm, 300 µm, 325 µm, or 350 µm), travel rates (1700–2250 mm/min), and flow rates (150–275 pulses/µL) were set to ensure adequate adhesion of the first layer to the print bed, consistent deposition of ink, and slight overlap between the first and second layers for each print. After printing, uncured prints were cured for 1.5–2 h under ambient conditions at 150 °C or 70 °C (depending on the catalyst).

### 2.3. Printability Determination

A few models correlating the rheological properties of the ink to the printed lattice structure have been proposed in the literature to predict printability. Examples include works by Smay et al. [14] (provided a condition for the storage modulus by correlating deflection in a strand to the distance between strands), M’Barki et al. [20] (evaluated the yield stress, print height, and capillary forces effects on printability), and Chan et al. [21] (provided an empirical relation between the equilibrium storage modulus, recovery of the storage modulus after shearing, and the yield stress). However, since none of these approaches were successful in predicting the printability for ink formulations made by this group, our team proposed an empirical model based on simple civil engineering problems that relate to the deflection of beams and supports under applied force. This engineering approach was used to correlate an ink’s rheology to the printability of a lattice structure based on the spacing ratio (a more detailed description of this model can be found in our previous work [19]). The rheology of the ink is accounted for by a variable referred to as the ink parameter (K*_ink_*) and defined by:(1)$${\;K}_{ink}={G^{\prime }}_{eq}\sigma_{y}{}^{-2}\rho_{ink}g,$$
where *ρ_ink_* represents the ink’s density, and *g* represents gravity. Along with the ink parameter, previous work by this group also proposed a lattice parameter (Ψ*_lattice_*), which is defined by: $\Psi_{lattice}=\eta^{4}d,$ where *η* is the spacing ratio, or the ratio of the center-to-center distance between adjacent strands to the diameter of the printed strands, and *d* is the diameter of the strut. These terms will be used to describe the printability of the inks in Section 3.2.

### 2.4. Material Characterization

Rheological properties of the inks created for this study were determined using a TA Discovery Series Hybrid Rheometer DHR-3 (TA Instruments, New Castle, DE, USA). Representative samples of each ink formulation were tested using a 25 mm cross-hatched parallel plate fixture geometry with strain sweeps conducted from 0.001% to 10% strain at an angular frequency of 10 rad/s and stress sweeps conducted from 10 to 10,000 Pa (or until the yield stress was reached) at an angular frequency of 10 rad/s. The yield stress (σ*_y_*) is determined either at the point where *G*′ starts to decrease or the point where *G*′ and *G*″ intercept each other; in this study, the yield stress was ascertained using the former method. Both the equilibrium storage moduli (*G’_eq_*) and yield stresses (σ*_y_*) of the inks were obtained using TA Instruments’ Trios software.

A confocal microscope (Keyence VHX-6000; Keyence Corporation, Osaka, Japan) was used to obtain images as well as determine the thicknesses of each printed pad. Overview and cross-section images were taken at 20× and 50× magnification, respectively, and thicknesses were determined using the Keyence analysis software. Elemental distributions within printed pads were determined via laser-induced breakdown spectroscopy (LIBS) using a Keyence VHX-7000 microscope with a Keyence EA-300 VHX Series element analysis system (Keyence Corporation). The elemental analyzer used a 10 µm spot size, and reported element concentrations (in wt%) were averaged from three analyses per spot.

An INSTRON 3343 Low-Force Testing System (INSTRON, Norwood, MA, USA) with BlueHill Universal software was used to perform uniaxial compression test on all printed samples. Each sample was subjected to 4 cycles of compression to a maximum stress of 0.4 MPa at a rate of 0.05 mm/s. The stress–strain curve for each print was determined by the final cycle. Because the displacement recorded by the BlueHill software was determined by the separation distance of the compression plates rather than the height of the sample, the displacement of each pad was corrected based on when the instrument first recorded force applied by the sample on the plates.

Thermal conductivity for printed pads was determined using a TA Fox 50 Heat Flow Meter (TA Instruments). During testing, each sample was pneumatically compressed to 60 psi between two thermally responsive plates while being heated from room temperature to 30 °C at 10 °C increments. Tests included two temperature regimes during which the upper and lower plates equilibrated to a 10 °C temperature variance (i.e., upper plate = 20 °C and lower plate = 10 °C). For these analyses, only the W-filled pads with <10% deviation in the desired printed thickness were tested as they were the only prints likely to be thermally conductive.

Finally, thermogravimetric analysis (TGA) was performed on printed pads representing each ink formulation using a TA Discovery Series TGA550 (TA Instruments). For these experiments, samples weighing ~5 mg were cut from representative pads and heated from 25 to 750 °C at a rate of 10 °C/min. During analysis, ultra-high purity nitrogen gas flowed across each sample at 40 mL/min, and the masses of the samples were tracked by the instrument. The onset of thermal degradation was defined as the point at which the sample lost 5% of its starting mass (*T_d5%_*); the decomposition temperature (*T_dMax_*) was defined as the temperature at which the derivative TGA (DTGA) curve was at a local maximum; and the final mass (*m_f_*) was defined as the residual mass of the sample at the end of the experiment. All of these values (*T_d5%_*, *T_dMax_*, and *m_f_*) were determined by the Trios software (TA Instruments) associated with the TGA.

## 3. Results and Discussion

### 3.1. Rheological Properties

As shown in Figure 1, the inks developed here exhibit shear-thinning yield stress and a rubbery plateau region, representing the strengths of physical cross-links between fillers and polymer chains and a given stress. For good printability, inks must display a rubbery plateau region and a yield stress easily achievable by extrusion during 3D printing. The best representation of these requirements in this work is the A300 ink, which shows a moderate *G*′*_eq_*, a well-defined rubbery plateau region, and a moderate yield stress, suggesting that this ink and inks with similar rheological properties—such as the 50 wt% W and 50 wt% WO_3_ inks—will work well as DIW feedstock (Figure 1 and Table 3). In contrast, very high or very low *G*′*_eq_* and yield stresses—such as those observed in the 70 wt% and 50 wt% Gd_2_O_3_ inks, respectively—will be much more difficult to print. Indeed, when attempting to print with the low-viscosity 50 wt% Gd_2_O_3_, TS-720, Emulsion, and A300 Emulsion inks, deposited layers immediately flowed and merged with lower layers creating thin, solid pads with few discernible struts rather than lattice structures with clearly distinguishable struts (Figure 2). Printing with the high-viscosity 70 wt% Gd_2_O_3_ ink proved to be impossible regardless of the nozzle size employed or pressure applied to extrude the ink (i.e., flow rate). These observations suggest some rheological bounding conditions for inks made from a PDMS resin base. For an ink to flow through the nozzle and be able to support itself after deposition (i.e., the struts formed do not flow and readily combine with neighboring struts), the *G′_eq_* and σ*_y_* must be within the following ranges: 42,655–2,825,820 Pa and 1990–11,455 Pa, respectively. Table 3 lists the rheological properties of various ink formulations developed in this work. These results demonstrate that inks based on siloxanes can be easily adapted to conform to various printing conditions, making these very attractive to additive manufacturing technology.

### 3.2. Printing Parameters and Printability

As seen in Table 2, the print settings used for each ink correlated well with the filler concentrations and subsequent rheological properties. Increases in filler concentration, *G*′, and yield stress led to necessary increases in the nozzle size, decreases in the travel rates, and increases in the flow rates used to print. While these adjustments in settings were necessary to print, they did not guarantee good printability, which refers to the ability of an ink to maintain its shape during the printing process without sagging or collapsing due to its own weight. Thus, the printability of the eleven inks formulated in this work was evaluated in terms of the lattice structure (FCT and SC), the diameter of the strands (250 µm, 350 µm, and 400 µm), and the spacing ratio. The center-to-center distances between adjacent strands for printed lattices in this study were 500 µm, 750 µm, 1000 µm, and 1500 µm, corresponding to spacing ratios of 2, 3, 4, and 6 for prints using 250 µm nozzles and spacing ratios of 1.22, 1.829, 2.439, and 3.659 for prints using 410 µm nozzles. Figure 3 shows examples of SC pads printed with increasing spacing ratios (Figure 3a–d), which results in increasing porosity and, depending on the rheology of the ink, the likelihood of strand sagging, or poor printability (Figure 3e,f).

In analogy to our previous work [19], variations on the printing parameters were considered in terms of the ratio of the center-to-center distance between adjacent strands to the diameter of the printed strands. This printing parameter is defined here as the spacing ratio (*η*). As the spacing ratio increases, so does the likelihood of the strands to collapse under their own weight, resulting in structures that are denser than initially predicted. Figure 4 shows the thickness deviation of each 3D printed structure as a function of the spacing ratio for a combination of ink formulations and lattice structures. Clearly, more data points fall outside the 10% and 20% thickness deviation range with increasing spacing ratio, which is a sign of reduced printability. When comparing FCT prints with SC prints, one can see that more FCT prints fall in the <10% deviation region than SC prints. This may be a result of the FCT structure allowing for fewer unsupported spans throughout the print compared to the SC structure, resulting in less sagging overall.

While most inks containing > 50 wt% filler necessitated the use of the larger 410 µm nozzles to print, one of these inks (60 wt% Gd_2_O_3_) was also able to be printed with the smaller 250 µm nozzle as well as the 410 µm nozzle (Table 2 and Figure 5). Because of this, the effect of nozzle size, or strut diameter, on thickness deviation as a function of spacing ratio could be investigated for this ink. In Figure 5, one can see that both FCT and SC prints created with the smaller 250 µm nozzle resulted in larger thickness deviations in the final prints. This suggests that the strut diameter is an important factor in determining whether an ink will have good printability in an FCT or SC geometry, where smaller strut diameters may lead to a higher probability of upper layers sagging and/or sinking into lower layers as is seen in Figure 6.

It is appealing to try to understand printability in terms of the ink density. Inks created in this study had densities ranging from 1.42 g/cm^3^ to 4.19 g/cm^3^ (Table 1), meaning if printability could be determined by ink density, there should be a clear trend in thickness deviation with increasing or decreasing density. However, as shown in Figure 7, this is not the case. For both FCT and SC structures, there is no correlation between thickness deviation as a function of print spacing ratio and ink density. For example, pads printed with the lowest density ink (the A300 ink represented by the lightest symbols) performed similarly to the densest printable ink (the 70 wt% W ink represented by the darkest symbols). Consequently, a more phenomenological approach is necessary to estimate printability.

To better understand the influence of ink rheology on printability, the K*_ink_* was determined (Table 4) and compared to the thickness deviation of the printed pads (Figure 8). Based on this data, it appears that the rheological properties of the ink alone are not able to predict printability as no clear trends emerge for FCT or SC prints.

The final potential predictor of DIW ink printability for FCT and SC lattice structures investigated in this study was the relationship between ink rheology and the lattice parameter, Ψ*_lattice_*. Figure 9 shows a plot of ink rheology (K*_ink_*) versus lattice parameters (Ψ*_lattice_*) with color-coded symbols representing pad thickness deviations for all inks printed in this study. When looking at all of the prints shown in Figure 9, there does not appear to be any correlation between printability (represented by the thickness deviations) and lattice parameters as a function of ink rheology. This is especially true for pads printed with gadolinium oxide- and tungsten oxide-filled inks. However, a trend informing printability does begin to emerge when prints made with A300 and tungsten-filled inks are singled out (Figure 10).

Figure 10 shows a subset of data points from Figure 9 representing prints made with A300 and tungsten-filled inks. Unlike the ceramic-filled inks (made with Gd_2_O_3_ and WO_3_), prints made with A300 and tungsten metal-filled inks show a trend toward increasing thickness deviation—or decreasing printability—with increasing Ψ*_lattice_* for each K*_ink_* value. While this trend may give one a starting point for determining printability in A300 and metal-filled inks, there is still no way to quantitatively demarcate K*_ink_* and Ψ*_lattice_* printability boundaries based on a given thickness deviation threshold (i.e., <10% thickness deviation).

### 3.3. Filler Composition and Printability

Strut diameter (or nozzle size) and spacing ratio appear to have the most influence on the thickness deviation, or printability, of DIW materials printed with FCT and SC lattice structures. However, the correlation between printability and ink characteristics such as rheology or density is less certain. While it is clear that an ink’s density or rheology alone cannot be used to predict printability (Figure 7 and Figure 8), there does appear to be a trend toward decreasing printability with increasing Ψ*_lattice_* for certain ink compositions including A300 and tungsten-filled (Figure 10). This implies the existence of a characteristic common to the silica- and metal-filled inks that distinguishes them from the ceramic-filled inks.

Looking at high-resolution microscope images of DIW pads printed with A300, 60 wt% W, 60 wt% WO_3_, and 60 wt% Gd_2_O_3_ inks, one can begin to see a few things that may explain why the printability of the silica- and metal-filled inks may be more predictable than the printability of the ceramic-filled inks. The first difference between the two groups of inks is that the silica- and metal-filled inks produce printed struts with very smooth, non-porous, reflective surfaces suggesting the complete encapsulation of fillers within the PDMS base (Figure 11a,b). On the other hand, the ceramic-filled inks seem to produce printed struts that are rougher and less reflective suggesting that these fillers break through the PDMS rather than being completely encapsulated by it (Figure 11d). This tearing of the polymers making up the base of the ink could explain some of the relative unpredictability of the printability of the ceramic-filled inks as the loss of strut coherence may lead to structural failures and printed pad thickness deviations outside of strut sagging due to gravity.

The second difference between prints made with the silica- and metal-filled inks and those made with the ceramic-filled inks is the distribution and appearance of the filler particles. In the silica-filled ink, one cannot distinguish silica particles from the PDMS base suggesting that the fumed silica filler is either completely bound to the PDMS or that the particles are simply impossible to see with microscopy. Based on the elemental analysis results in Figure 12a showing areas with high silicon concentration (spots 1, 4, and 5; likely representing relatively pure fumed silica) surrounded by a higher number of areas with high oxygen and low silicon concentrations (spots 2, 3, and 6–9; likely representing PDMS), it appears that a combination of these is correct. For the tungsten metal-filled ink, the filler is much easier to identify compared to the silica-filled ink, but it is still difficult to pick out individual particles suggesting that the particle size distribution is fairly homogenous and that the filler is evenly distributed throughout the resin (Figure 11b). This is confirmed by the elemental maps shown in Figure 12b, which exhibit similar amounts of tungsten mixed with carbon, silicon, oxygen, and hydrogen (representing a mixture of PDMS, fumed silica, and tungsten metal powder) in eight out of the nine LIBS spots (spots 1 and 3–9) and only one spot of relatively high tungsten concentration (spot 2; likely representing a pure tungsten particle). Those particles that can be distinguished individually appear to have fairly rounded edges (Figure 11b and Figure 12b).

In contrast to the prints made with silica- and metal-filled inks, prints made with tungsten oxide-filled inks have easily distinguished, sharp particles in varying sizes, suggesting a more heterogenous mixture of filler and PDMS in the resin (Figure 11c and Figure 12c). This is confirmed by the elemental distributions displayed in Figure 12c, which show distinct high-concentration areas of silicon (spot 1), tungsten (spots 4–5), and carbon (spots 3, 6–7, and 9) mixed with oxygen and hydrogen, likely representing fumed silica, tungsten oxide, and PDMS, respectively. Only spots 2 and 8 (Figure 12c) show a more even mixture of all components of the tungsten-oxide ink. When looking at the images of prints made with gadolinium oxide-filled ink, particles appear to be very homogenous and evenly distributed throughout the resin (Figure 11d). While individual particles are difficult to distinguish, the granular appearance of the printed structure suggests that the gadolinium oxide filler is relatively rough compared to the fumed silica or tungsten metal. The elemental maps in Figure 12d show a very homogenous mixture of gadolinium and silicon, though no other elements were detected, suggesting that the signals for elements representing the PDMS resin were overwhelmed by those representing the gadolinium oxide and fumed silica fillers. Based on these findings, rough or sharp filler particles, heterogenous filler particle sizes, and/or suboptimal polymer-filler interactions may contribute to poor and/or unpredictable ink printability.

### 3.4. Thermomechanical Characterization

One of the main reasons to include a lattice structure in a printed part is to control the mechanical compliance of the part. In previous work by this group [19], the positive relationship between increasing porosity (achieved by increasing the center-to-center distance between printed struts) and increasing compressive strain was established. Additionally, the differences in the behavior of FCT versus SC structures under compressive and shear stress has also been examined in previous studies [13]. In this study, the relationship between compressive strain and increasing filler concentrations in addition to FCT versus SC print geometries was investigated by measuring the compressive strain for FCT (Figure 13a) and SC (Figure 13b) prints with 500 µm center-to-center distance between struts made with each of the successfully printed inks. The maximum compressive strain (corresponding to a maximum compressive stress of 0.4 MPa) for each printed pad is shown in Table 5. Based on these results, the maximum compressive strain decreases with increasing filler concentration and is greater for prints with FCT geometries compared to those with SC geometries, suggesting that higher filler loadings and SC geometries lead to stiffer printed parts (Figure 13 and Table 5).

Another potentially tunable characteristic of DIW printed parts is thermal conductivity. Because metals, in general, have better conductivity than silicones and ceramics, thermal conductivity was only investigated for printed pads made with tungsten metal-filled inks. To determine the effects of filler concentration and lattice geometry on thermal conductivity of the tungsten metal-filled prints, pads with the same center-to-center distance between printed struts (500 µm) were analyzed for each filler concentration (50 wt%, 60 wt%, or 70 wt%) and lattice geometry (FCT or SC). In Figure 14, one observes an increase in thermal conductivity with increasing filler concentration as well as a trend of greater conductivity in prints with SC lattice geometries than prints with FCT lattice geometries. The increase in thermal conductivity with increasing metal concentration is intuitive, but the higher thermal conductivity of SC prints compared to FCT prints is not.

The instrument that was used to measure thermal conductivity applies a pressure of 60 psi to printed parts during analysis, which roughly corresponds to the maximum compressive stress applied to printed parts during compression analyses (0.4 MPa). Because of this, the relationship between the conductivity and the maximum compressive strain of the pads analyzed in Figure 14 was investigated (Figure 15). This analysis shows a *negative* correlation between thermal conductivity and maximum compressive strain, which contradicts previous findings [19] and suggests that another factor may be contributing. Conductivity heavily depends on the connectedness of the path from source to sink, which is why more compressed structures showed higher conductivity in [19]. However, in an SC lattice, there is a more direct path from source to sink compared to FCT lattice structures where conductivity is inherently obstructed by more air voids regardless of the level of compression (Figure 16). Therefore, with the same metal filler concentration, an SC printed lattice will be more thermally conductive than an FCT printed lattice.

The final thermomechanical analysis performed on DIW pads printed in this study was TGA. Figure 17 shows the TGA curves for representative printed samples of all ink formulations able to be printed. This figure, as well as Table 6, illustrate that the onset of thermal degradation (*T_d5%_*) for A300 printed pads is significantly lower than that of the metal- and ceramic-filled pads and that the *T_d5%_* and final mass (*m_f_*) of metal- and ceramic-filled pads increase with increasing filler concentration, confirming that the addition of fillers other than fumed silica to DIW inks increases the thermal stability of the ink as reported in [19]. Additionally, except for the 60 wt% pads, the metal-filled inks appear to be more thermally stable (lower *T_d5%_* and higher *m_f_* values) than their ceramic-filled counterparts (Figure 17 and Table 6).

## 4. Conclusions

There are many different knobs one can turn to customize the mechanical and thermal properties of parts printed with DIW inks. These options include augmenting functionality by adding fillers, modifying mechanical compliance through lattice geometries and spacings, and adjusting thermal stability and conductivity by varying filler concentrations and controlling source-to-sink pathways through the printed part. However, before turning these knobs, one must know how these variations will affect the printability of the ink and the intended structure. In this study, DIW inks containing a variety of functional fillers of varying concentrations were created and printed into pads with two different lattice geometries: FCT and SC. For each printed pad, the filler types, filler concentrations, ink densities, ink rheologies, strut diameters, spacing ratios, lattice geometries, and particle characteristics were compared to thickness deviations in the final printed part to determine if there were any predictors of printability. Based on the results of this work, strut diameter (or nozzle size) and spacing ratio appear to have the most influence on the thickness deviation, or printability, of DIW materials. Additionally, silica- and metal-filled inks showed a trend toward decreasing printability with increasing lattice parameter (Ψ*_lattice_*) values, while ceramic-filled inks did not have any clear trends in printability. Upon further inspection of these prints, the fillers in the first groups seemed to be completely encapsulated by the PDMS resin leading to more predictable printability based on strut sagging, while the ceramic fillers tore through the resin leading to a loss of strut coherence and print thickness deviations outside of gravity-based strut sagging.

When investigating the thermomechanical properties of the inks created in this study, it was determined that higher filler loadings and SC geometries led to stiffer printed parts than lower loadings and FCT geometries. Thermal conductivity in pads printed with metal-filled inks was found to be greater for pads printed with SC geometries than those printed with FCT geometries due to the more congruent source-to-sink heat pathway available in SC lattices. Finally, increasing filler concentrations in the DIW inks investigated lead to increases in the thermal stability of the printed parts, and metal-filled inks appeared to produce more thermally stable prints than their ceramic-filled counterparts. Overall, this work provides more insight into the tunability of thermomechanical properties and printability of custom inks and DIW printed parts.

## Figures and Tables

**Figure 1 polymers-14-04661-f001:**
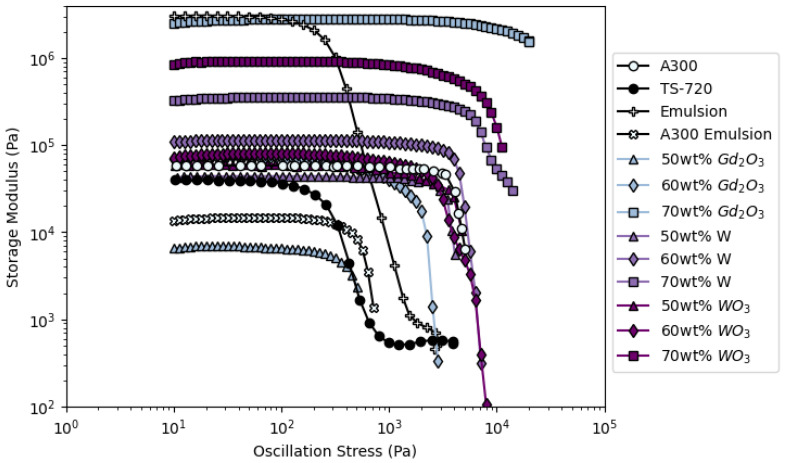
Rheological analyses results showing the storage modulus as a function of oscillation stress for representative samples of the DIW inks created in this study.

**Figure 2 polymers-14-04661-f002:**
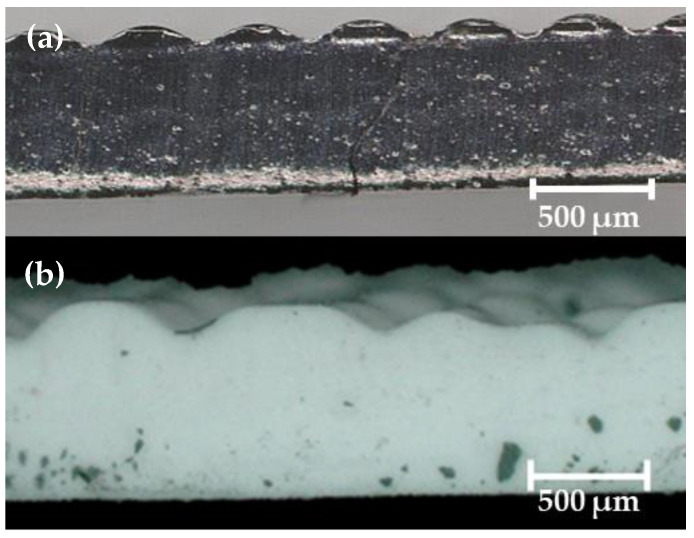
Side view of FCT pads printed with: (**a**) TS-720 and (**b**) 50 wt% Gd_2_O_3_ DIW inks.

**Figure 3 polymers-14-04661-f003:**
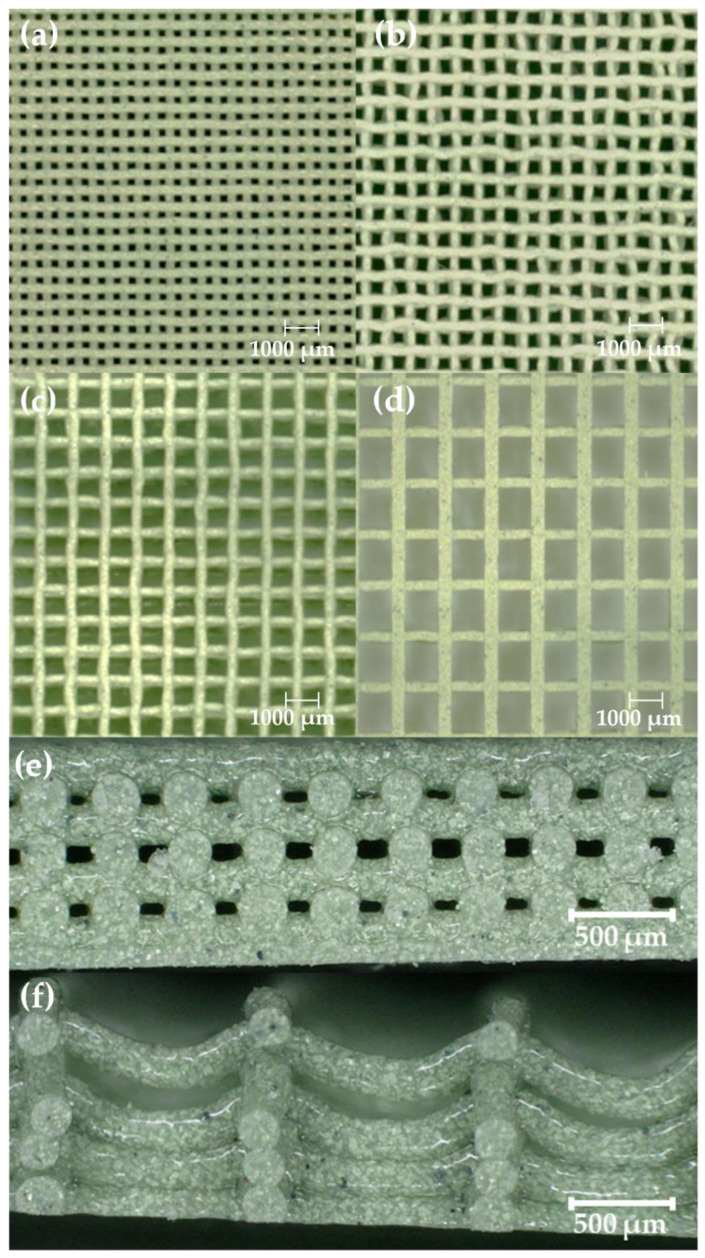
Top view microscopy images of SC parts printed with 50 wt% WO_3_ ink using a 250 µm nozzle at: (**a**) 500 µm or *η* = 2; (**b**) 750 µm or *η* = 3; (**c**) 1000 µm or *η* = 4; and (**d**) 1500 µm or *η* = 6. Side view microscopy images (50×) of SC parts printed with 50 wt% WO_3_ ink using a 250 µm nozzle showing stable vs. sagging struts in prints with spacing: (**e**) 500 µm or *η* = 2; and (**f**) 1500 µm or *η* = 6.

**Figure 4 polymers-14-04661-f004:**
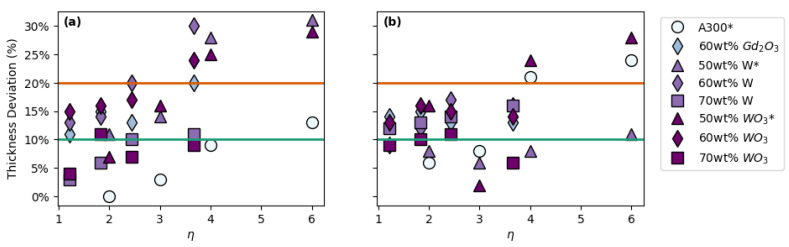
Overall thickness deviation vs. spacing ratio (*η*) for prints with: (**a**) FCT geometries and (**b**) SC geometries. * Data in plot (**a**) from Brounstein et al. [19].

**Figure 5 polymers-14-04661-f005:**
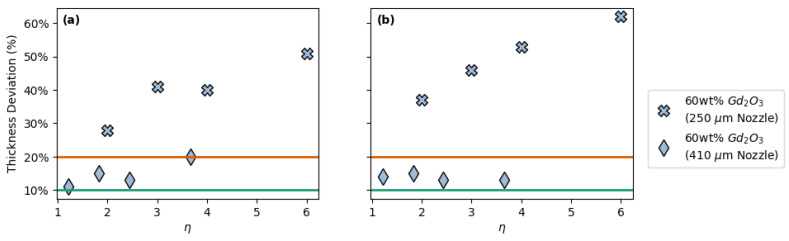
Overall thickness deviation vs. spacing ratio (*η*) for 60 wt% Gd_2_O_3_ prints using 250 µm and 410 µm nozzles with: (**a**) FCT geometries and (**b**) SC geometries.

**Figure 6 polymers-14-04661-f006:**
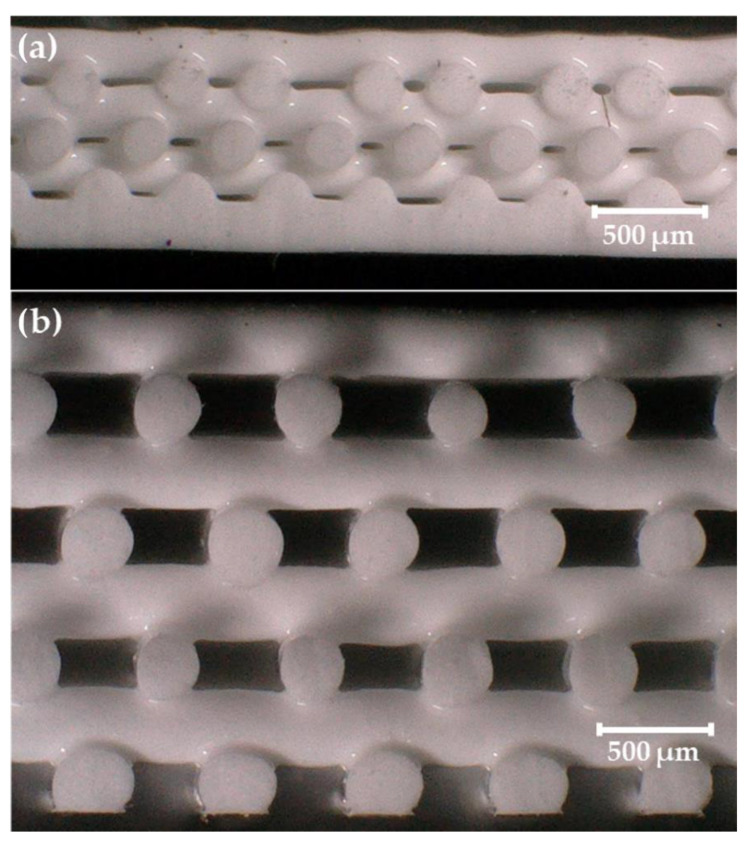
Side view microscope images of 60 wt% Gd_2_O_3_ FCT prints using: (**a**) a 250 µm nozzle with 500 µm center-to-center strut distance (*η* = 2) and (**b**) a 410 µm nozzle with 750 µm center-to-center strut distance (*η* = 1.829).

**Figure 7 polymers-14-04661-f007:**
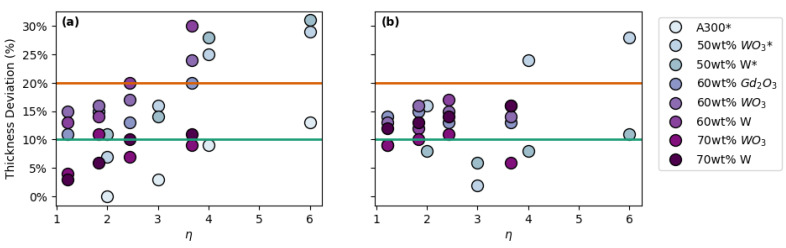
Relating print thickness deviation to ink density *(ρ_ink_*) and spacing ratios (*η*) for: (**a**) FCT prints and (**b**) SC prints. Darker colors represent higher ink densities, and lighter colors represent lower ink densities. * Data in plot (**a**) from Brounstein et al. [19].

**Figure 8 polymers-14-04661-f008:**
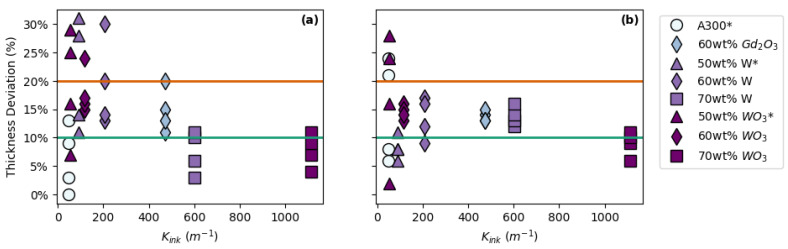
Relating print thickness deviation to ink rheology (K*_ink_*) for: (**a**) FCT prints and (**b**) SC prints. * Data in plot (**a**) from Brounstein et al. [19].

**Figure 9 polymers-14-04661-f009:**
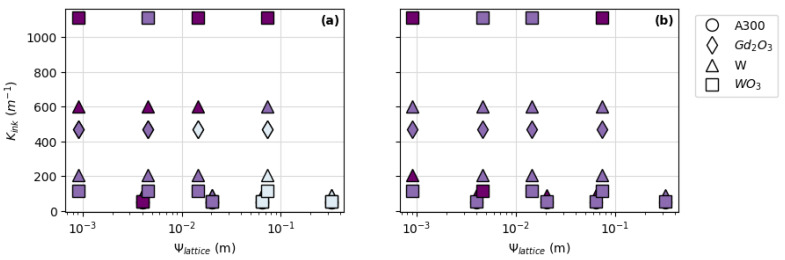
Ink rheology (K*_ink_*) vs. lattice parameter (Ψ*_lattice_*) for: (**a**) FCT prints and (**b**) SC prints. The darkest symbols represent pads with <10% thickness deviation; the second-darkest symbols represent pads with <20% thickness deviation; and the lightest symbols represent pads with >20% thickness deviation.

**Figure 10 polymers-14-04661-f010:**
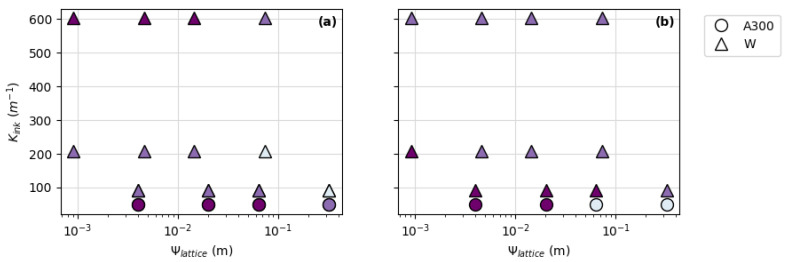
Subset of data points from Figure 9 showing ink rheology (K*_ink_*) vs. lattice parameter (Ψ*_lattice_*) for A300 and tungsten-filled: (**a**) FCT prints and (**b**) SC prints. The darkest symbols represent pads with <10% thickness deviation; the second-darkest symbols represent pads with <20% thickness deviation; and the lightest symbols represent pads with >20% thickness deviation.

**Figure 11 polymers-14-04661-f011:**
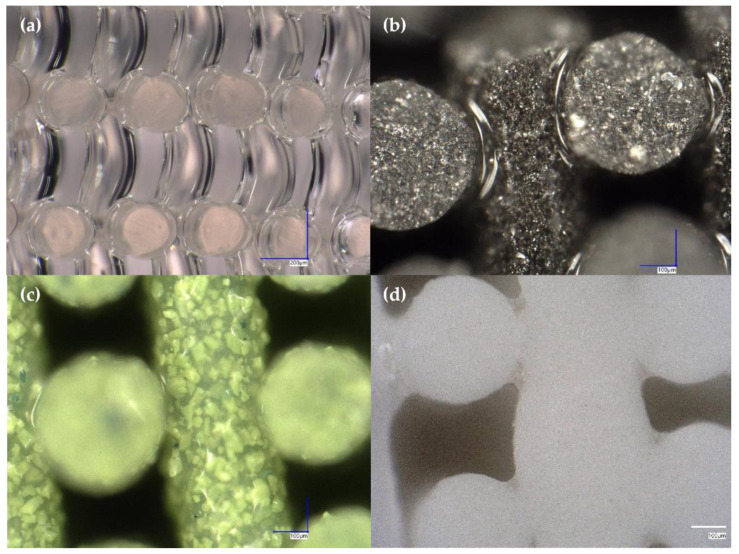
Microscope images of DIW prints made with: (**a**) A300 ink, (**b**) 60 wt% W ink, (**c**) 60 wt% WO_3_ ink, and (**d**) 60 wt% Gd_2_O_3_ ink.

**Figure 12 polymers-14-04661-f012:**
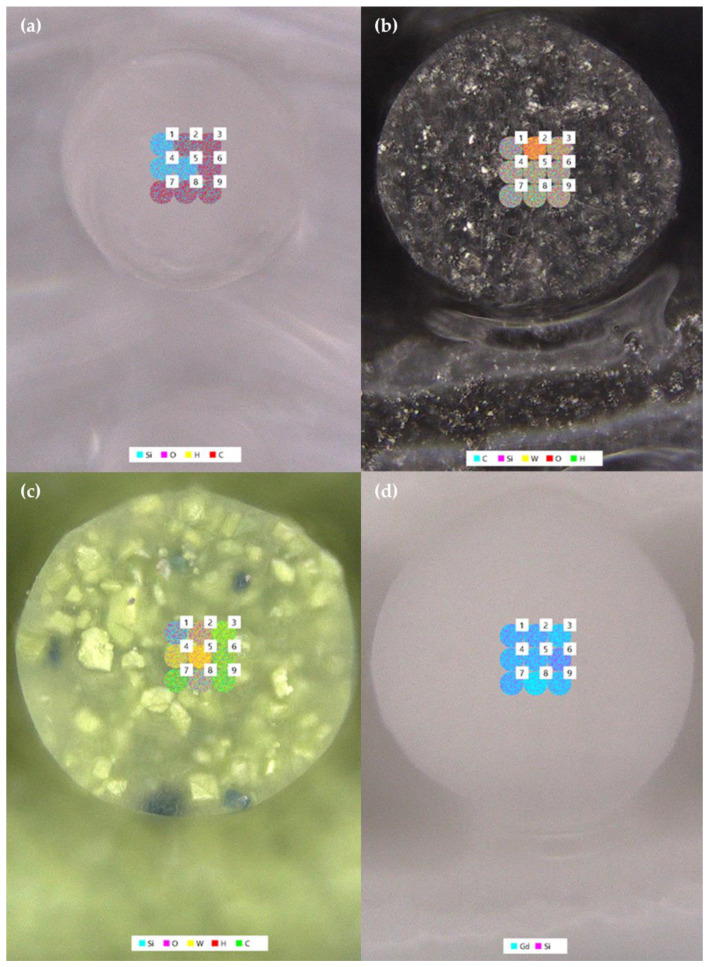
Microscope images and elemental maps of DIW prints made with: (**a**) A300 ink, (**b**) 60 wt% W ink, (**c**) 60 wt% WO_3_ ink, and (**d**) 60 wt% Gd_2_O_3_ ink.

**Figure 13 polymers-14-04661-f013:**
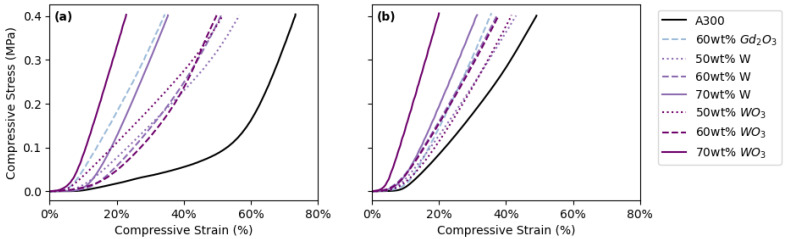
Compressive stress–strain curves for DIW pads with 500 µm center-to-center distance between struts printed with: (**a**) FCT lattice geometry and (**b**) SC lattice geometry.

**Figure 14 polymers-14-04661-f014:**
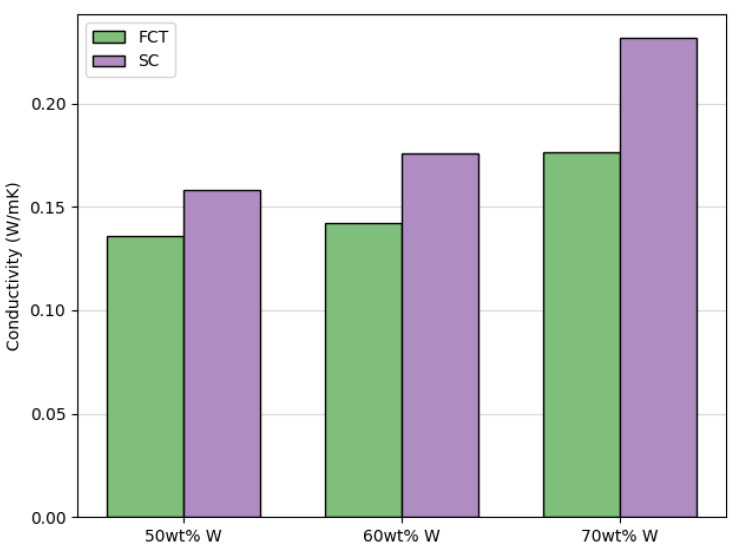
Thermal conductivity of tungsten-filled DIW pads printed with FCT and SC lattice structures with 500 µm center-to-center strut spacing.

**Figure 15 polymers-14-04661-f015:**
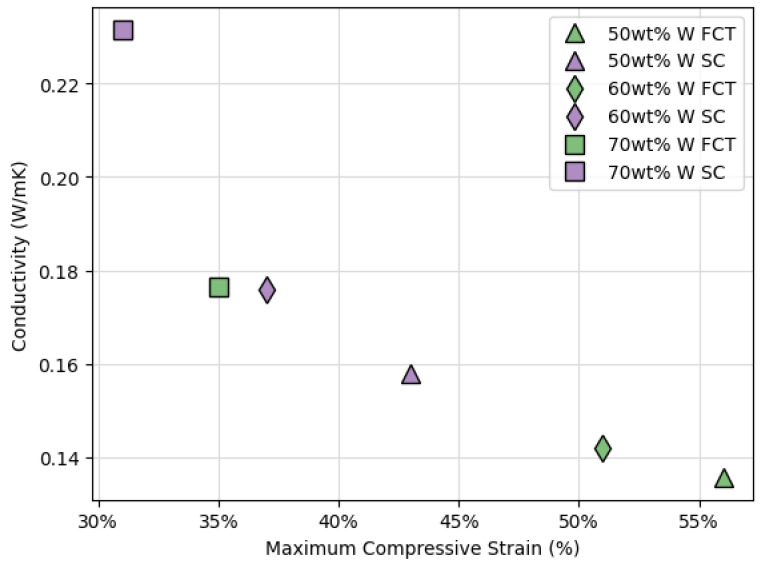
Thermal conductivity versus compressive strain of tungsten-filled DIW pads printed with FCT and SC lattice structures and 500 µm center-to-center strut spacing.

**Figure 16 polymers-14-04661-f016:**
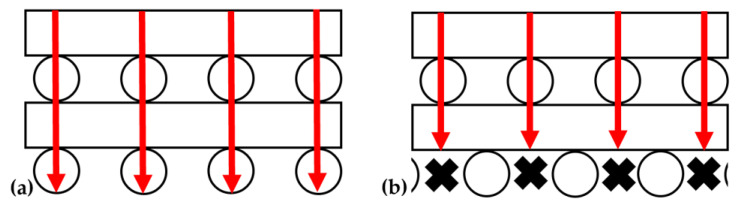
Schematic of thermal conductivity pathways in pads with: (**a**) SC lattice structure and (**b**) FCT lattice structure.

**Figure 17 polymers-14-04661-f017:**
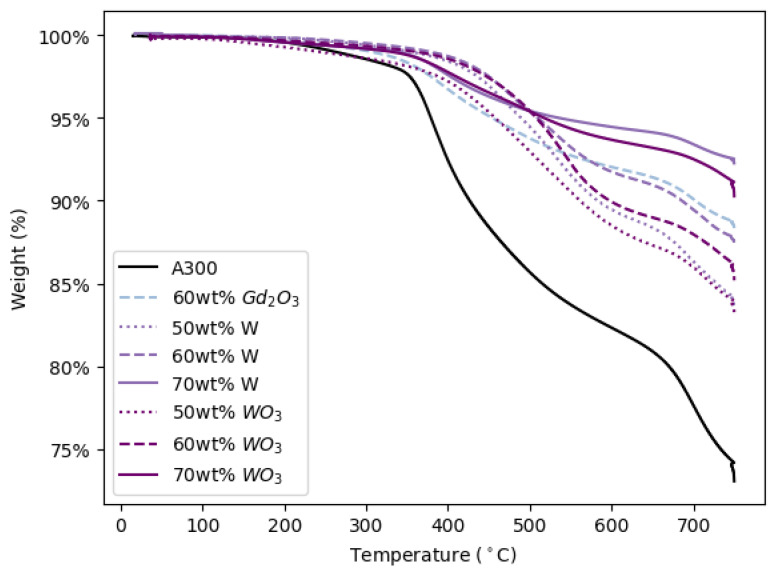
TGA curves for representative samples of each 3D printed formulation.

**Table 1 polymers-14-04661-t001:** Weight percent fillers and densities for ink formulations used in this study.

Formulation	Weight % Metal/ Ceramic Filler	Weight % Fumed Silica	Density (g/cm^3^)
A300	0	10	1.42
TS-720	0	15	1.45
Emulsion	0	0	1.32
A300 Emulsion	0	10	1.34
W	50	4.5	2.66
	60	4.5	3.18
	70	4.5	4.19
WO_3_	50	4.5	2.36
	60	4.5	2.74
	70	4.5	3.28
Gd_2_O_3_	50	4.5	1.85
	60	4.5	2.67
	70	4.5	3.23

**Table 2 polymers-14-04661-t002:** Print settings for ink formulations used in this study.

Formulation	Nozzle Size (µm)	Layer Height (µm)	Travel Rate (mm/min)	Flow Rate (pulses/µm)
A300	250	225	2240	150
TS-720	350	325	N/A **	N/A **
Emulsion	350	225	N/A **	N/A **
A300 Emulsion	250	225	N/A **	N/A **
50 wt% W	250	225	2240	195
60 wt% W	410	350	2225	235
70 wt% W	410	350	1980	260
50 wt% WO_3_	250	225	2250	150
60 wt% WO_3_	410	325	2200/2100 *	180
70 wt% WO_3_	410	300	1700/1600 *	275
50 wt% Gd_2_O_3_	250	225	N/A **	N/A **
60 wt% Gd_2_O_3_ ^†^	250/410	225/325	2250/2240	120/225
70 wt% Gd_2_O_3_	410	350	N/A **	N/A **

* For these inks, the travel rate on the left was used for the first layer, and the travel rate on the right was used for all subsequent layers. ^†^ The 60 wt% Gd_2_O_3_ ink was printed with both a 250 µm nozzle and a 410 µm nozzle; values on the left represent print settings using a 250 µm nozzle, and values on the right represent print settings using a 410 µm nozzle. ** These inks do not have a travel rate or flow rate listed because successful rates could not be determined.

**Table 3 polymers-14-04661-t003:** Rheological properties of a representative sample of each ink formulation created in this study.

Formulation	Weight % Metal/ Ceramic Filler	Equilibrium Storage Modulus (*G’_eq_*) (Pa)	Yield Stress (σ*_y_*) (Pa)
A300	0	55,525	1990
TS-720	0	39,783	207
Emulsion	0	3,784,860	159
A300 Emulsion	0	14,930	448
W	50	42,655	3495
	60	114,374	4150
	70	358,137	4283
WO_3_	50	57,405	5021
	60	79,908	4283
	70	923,592	5165
Gd_2_O_3_	50	6980	320
	60	75,327	2043
	70	2,825,820	11,455

**Table 4 polymers-14-04661-t004:** Rheological properties of a representative sample of each ink formulation used in this study.

Formulation	Weight % Metal/ Ceramic Filler	K*_ink_* (m^−1^)
SiO_2_	0	49
TS-720	0	13,206
Emulsion	0	1,938,646
A300 Emulsion	0	978
W	50	91
	60	207
	70	602
WO_3_	50	53
	60	117
	70	1113
Gd_2_O_3_	50	1237
	60	473
	70	682

**Table 5 polymers-14-04661-t005:** Maximum compressive strain (displacement) at 0.4 MPa for DIW pads printed at 500 µm center-to-center distance spacing between struts.

Formulation	Weight % Metal/ Ceramic Filler	Lattice Geometry	Max Compressive Strain (%)
A300	0	FCT	73
		SC	49
Gd_2_O_3_	60	FCT	34
		SC	35
W	50	FCT	56
		SC	43
	60	FCT	51
		SC	37
	70	FCT	35
		SC	31
WO_3_	50	FCT	52
		SC	42
	60	FCT	50
		SC	38
	70	FCT	23
		SC	20

**Table 6 polymers-14-04661-t006:** Thermal stability properties of a representative sample of each 3D printed formulation.

Formulation	T*_d5%_* (°C)	*m_f_* (%)
A300	379	73
60 wt% Gd_2_O_3_	453	88
50 wt% W	489	84
60 wt% W	509	88
70 wt% W	534	92
50 wt% WO_3_	458	83
60 wt% WO_3_	508	85
70 wt% WO_3_	516	90

## Data Availability

The authors confirm that the data supporting the findings of this study are available within the article.

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
