# Peer review of "Balancing Functionality and Printability: High-Loading Polymer Resins for Direct Ink Writing"

_polymers, 2022, doi:10.3390/polym14214661_

Round 1

Reviewer 1 Report

Dear Authors

This is a great study about DIW based on printable inks with high loading of ceramic or metal particles. I hope that a lot of applications are envisioned already.

The study is well described and organized in suitable sections. Materials and methods are well represented in an extensive way and consequently suitable long description is given for the Result/discussion section.

The study is nice to read because in this study, basic topics related to study are covered and someone can learn from it.

The conclusion part is well-established ad consequently is a summary of the obtained results.

Some suggestions for improving this work could be found in the attached version of the manuscript. Thank you.

Reviewer 2 Report

In this manuscript, the authors investigated the effects of filler composition/concentration, printing parameters, and lattice structure on the printability of new polysiloxane inks with high concentrations of metallic and ceramic fillers as well as emulsions. I think the authors give a detailed assessment of printing parameters and characterization. There are some concerns about experimental designs and outcome discussion. I would like to recommend a major revision before the manuscript can be accepted.

Comments:

1. Please give a brief description of critical results in abstract.

2. Introduction: The authors should point out the current challenges in DIW, for example: tuning the ink's rheological properties (e.g, shear-thinning behaviors) to enable flow through a nozzle followed by thickening after extrusion for good shape retention. Also, the authors should give more recent works or reviews that are related to the strategies to solve these problems. (for example: doi.org/10.1002/adma.202108855; doi.org/10.1016/j.cej.2021.128541; doi.org/10.1016/j.matdes.2020.109337)

3. Emulsions used in the manuscript were over 50%, which could be beneficial to the mechanical strength after printing. But the aggregation or ink clogging may be big concerns. The authors should offer some SEM or EDS analysis to discuss these common issues.

4. ‘Emulsion ink, water was added to the resin to lower viscosity and promote better flow’ Will these water cause phase separation? Why not use organic solvent?

5. How soon will the crosslinking occur after printing or curing?

6. How about the rheological properties under temperature variation? Will 150°c cause serve collision during curing?

7. Figure 11a, high-resolution microscope image of DIW pad (Si only ink) need to be retaken.

8. Please simplify the conclusion session.

Round 2

Reviewer 2 Report

The authors have addressed all my concerns.